# Pancreatic Pathological Changes in Murine Toxoplasmosis and Possible Association with Diabetes Mellitus

**DOI:** 10.3390/biomedicines11010018

**Published:** 2022-12-22

**Authors:** Asmaa M. El-kady, Amal M. Alzahrani, Hayam Elshazly, Eman Abdullah Alshehri, Majed H. Wakid, Hattan S. Gattan, Wafa Abdullah I. Al-Megrin, Mashael S. Alfaifi, Khalil Mohamed, Waheeb Alharbi, Hatem A. Elshabrawy, Salwa S. Younis

**Affiliations:** 1Department of Medical Parasitology, Faculty of Medicine, South Valley University, Qena 83523, Egypt; 2Department of Biology, Faculty of Sciences & Arts in Almandaq, Al Baha University, Al Baha 65779, Saudi Arabia; 3Department of Biology, Faculty of Sciences-Scientific Departments, Qassim University, Buraidah 52571, Saudi Arabia; 4Department of Zoology, Faculty of Science, Beni-Suef University, Beni Suef 62521, Egypt; 5Department of Zoology, College of Science, King Saud University, Riyadh 11362, Saudi Arabia; 6Department of Medical Laboratory Technology, Faculty of Applied Medical Sciences, King Abdulaziz University, Jeddah 21589, Saudi Arabia; 7Special Infectious Agents Unit, King Fahd Medical Research Center, Jeddah 21589, Saudi Arabia; 8Department of Biology, College of Science, Princess Nourah bint Abdulrahman University, P.O. Box 84428, Riyadh 11671, Saudi Arabia; 9Department of Epidemiology, Faculty of Public Health and Health Informatics, Umm Al-Qura University, Mecca 21961, Saudi Arabia; 10Department of Physiology, Faculty of Medicine, Umm Al-Qura University, Mecca 21961, Saudi Arabia; 11Department of Molecular and Cellular Biology, College of Osteopathic Medicine, Sam Houston State University, Conroe, TX 77304, USA; 12Departments of Medical Parasitology, Faculty of Medicine, Alexandria University, Alexandria 21131, Egypt

**Keywords:** *Toxoplasma gondii*, caspase-3, apoptosis, CD8, insulin, islets of Langerhans, diabetes, T1DM

## Abstract

Background: Previous studies have reported involvement of *Toxoplasma gondii* (*T. gondii)* infections in the pathogenesis of some autoimmune diseases, such as polymyositis, rheumatoid arthritis, autoimmune thyroiditis, and Crohn’s disease. However, data on the association between *T. gondii* infections and Type 1 diabetes mellitus (T1DM) are still controversial. Therefore, in the present study, we aimed to investigate the pancreatic pathological changes in mouse models with acute and chronic toxoplasmosis and their association with T1DM. Materials and Methods: Three groups (10 mice each) of male Swiss Albino mice were used. One group of mice was left uninfected, whereas the second and third groups were infected with the acute virulent *T. gondii* RH strain and the chronic less virulent Me49 *T. gondii* strain, respectively. *T. gondii*-induced pancreatic pathological changes were evaluated by histopathological examination of pancreatic tissues. Moreover, the expression of insulin, levels of caspase-3, and the pancreatic infiltration of CD8^+^ T cells were evaluated using immunohistochemical staining. Results: Pancreatic tissues of *T. gondii*-infected animals showed significant pathological alterations and variable degrees of insulitis. Mice with acute toxoplasmosis exhibited marked enlargement and reduced numbers of islets of Langerhans. However, mice with chronic toxoplasmosis showed considerable reduction in size and number of islets of Langerhans. Moreover, insulin staining revealed significant reduction in β cell numbers, whereas caspase-3 staining showed induced apoptosis in islets of Langerhans of acute toxoplasmosis and chronic toxoplasmosis mice compared to uninfected mice. We detected infiltration of CD8^+^ T cells only in islets of Langerhans of mice with chronic toxoplasmosis. Conclusions: Acute and chronic toxoplasmosis mice displayed marked pancreatic pathological changes with reduced numbers of islets of Langerhans and insulin-producing-β cells. Since damage of β cells of islets of Langerhans is associated with the development of T1DM, our findings may support a link between *T. gondii* infections and the development of T1DM.

## 1. Introduction

*Toxoplasma gondii* (*T. gondii*) is an obligate intracellular opportunistic parasite that infects almost all mammals, including humans [1,2]. Infections with *T. gondii* are usually asymptomatic in immunocompetent individuals; however, *T. gondii* can cause serious disease immunocompromised individuals and pregnant women [3,4,5,6]. Humans can acquire infection through ingestion of tissue cysts in meat, food and water contaminated with oocysts from infected cats, and congenitally from infected mothers [7].

It has been well documented that Th1 cell-mediated and humoral immune responses develop following *T. gondii* infection [8]. Although these immune responses are required for the host defense against *T. gondii*, excessive inflammatory response damages the host tissues [9]. Additionally, *T. gondii* infection could result in autoantibodies production, which may further potentiate tissue damage [10].

Previous clinical studies have associated anti-*Toxoplasma* antibodies with several autoimmune diseases (AID) including polymyositis [11], rheumatoid arthritis (RA) [12,13,14], autoimmune thyroid diseases [11,15,16], Crohn’s disease [17], anti-phospholipid syndrome [18], Wegener’s granulomatosis [19] and autoimmune bullous diseases [20]. However, experimental studies reported conflicting findings on the association between AID and *T. gondii* infection [21,22,23].

Type 1 diabetes mellitus (T1DM) is an autoimmune disorder that results from the destruction of β cells of islets of Langerhans in the pancreas by antigen-specific T lymphocytes [24]. To date, 0.3% of the US population have type 1 diabetes (1.6 million per 330 million US residents) [25]. The pathological characteristic of T1DM is insulitis, which is defined by autoreactive CD4+ and CD8+ T-cell infiltration of insulin-producing-β cells of islets of Langerhans [26,27,28]. Similar to other autoimmune diseases, genetic and environmental factors, such as infectious agents, are involved in the development of T1DM [27,29].

Data on the association between *T. gondii* infection and T1DM are still controversial [30]. Several studies have reported that diabetic patients have a high seroprevalence of toxoplasmosis antibodies [30,31,32,33,34,35,36]. On the other hand, the ability of *T. gondii* to infiltrate and proliferate inside pancreatic cells has been linked to an increased risk of developing diabetes [36,37].

In the present study, we evaluated the pancreatic pathological changes in mice infected with acute virulent and chronic less virulent *T. gondii* strains. Our findings demonstrated insulitis and marked reduction in numbers of β cells of islets of Langerhans in pancreatic tissues of infected animals associated with apoptotic cell death. Therefore, our data may provide a possible link between *T. gondii* infections and development of T1DM.

## 2. Materials and Methods

### 2.1. Animal Experiment

The present study was conducted at the Department of Medical Parasitology, Faculty of Medicine, Alexandria University, Egypt.

Four to six week old laboratory bred male Swiss Albino mice, weighing 20–25 g, were used in our experiment. Mice were housed in well-ventilated cages and were provided with water and fed on regular pellet meals. All mice groups were kept under strict light cycles (12 h light/12 h dark cycle) and a temperature of 25 ± 2 °C). Stool examination was performed for three successive days to rule out any parasitic infections using the formol-ether concentration method [38] and modified Ziehl–Neelsen technique [39]. Mice were divided into three groups (10 mice each), which included the uninfected control group, acute toxoplasmosis group (infected intraperitoneally with 1 × 10^4^ tachyzoites of the virulent RH HXGPRT (-) strain of *T. gondii* tachyzoites/mouse, and chronic toxoplasmosis group (infected orally with 10 cysts/mouse of the Me49 less virulent strain of *T. gondii)*. Mice in the acute toxoplasmosis group were sacrificed 5 days post-infection (PI) [36], whereas mice in the chronic toxoplasmosis group were sacrificed 60 days PI [36].

#### 2.1.1. Infection with Me49 Strain of *T. gondii*

Ten mice were orally infected with cysts of Me49 strain (10 cysts/mouse) obtained from brain homogenate of infected mice, 8 weeks PI [40,41]. Briefly, the brain homogenate was prepared by homogenizing each infected brain in 1 mL saline using a tissue homogenizer (Wheaton, IL, USA). Cysts were microscopically counted using hemocytometer under 400× magnification. The brain suspension was then diluted to a concentration of 100 cysts/mL and each mouse was infected with 0.1 mL containing 10 cysts [41].

#### 2.1.2. Infection with Virulent RH Strain of *T. gondii*

Tachyzoites of the virulent RH strain of *T. gondii* were obtained by serial intraperitoneal passages in Swiss Albino mice. Briefly, tachyzoites were obtained by flushing the peritoneal cavity with Phosphate Buffered Saline (PBS) 5 days post infection [42]. The peritoneal fluid was centrifuged at 200× *g* for 5 min at room temperature, to remove peritoneal cells and cellular debris. The supernatant was then centrifuged for 10 min at 800× *g* [43], and the residue containing the tachyzoites was counted by hemocytometer then diluted in PBS to the concentration of 1 × 10^5^/mL. Each mouse was infected with 1 × 10^4^ in 0.1 mL PBS [44].

### 2.2. Histopathological Examination

#### 2.2.1. Hematoxylin and Eosin Staining

Pancreas was isolated from animals of all groups, fixed in 10% formalin, dehydrated in ascending concentrations of ethanol, and embedded in paraffin. Sections of 3 µm thickness were prepared and stained with hematoxylin and eosin (H&E) stain and then images were taken and examined by an independent pathologist. ImageJ scanner and viewer software were used to scan slides and process images (LOCI, University of Wisconsin, Madison, WI, US).

Pancreatic tissue sections were examined for the degree of necrosis and inflammatory cell infiltration. In addition, the size, number, and the morphology of islets of Langerhans/mouse were examined in 3 random high power fields (HPF; 400x) and mean was calculated/group. In addition, the presence of insulitis was evaluated and graded on a scale 0–4 as previously described: islets devoid of any mononuclear cells = 0; minimum focal islet infiltrate = 1+; peri-islet infiltrate of <25% of islet circumference = 2+; peri-islet infiltration and <50% intra-islet area = 3+; intra-islet infiltration >50% of islet area = 4+ [45].

#### 2.2.2. Immunohistochemistry

Pancreatic tissues were examined for expression of insulin, levels of caspase-3, and infiltration of CD8 T cells. Briefly, 4 µm thick paraffin pancreatic tissue sections were de-paraffinized in xylene for 20 min, rehydrated with descending ethanol concentrations and then rinsed in distilled water. Sections were placed in citrate buffer (pH 6.0) and heated in microwave for epitope retrieval. Endogenous peroxidases were then blocked by incubating in 0.6% H_2_O_2_ for 10 min. Tissues sections were washed twice with PBS then treated with superblock and incubated overnight at room temperature with the following primary antibodies: Anti-insulin rabbit monoclonal antibody (Catalog no.: A19066, ABclonal, Woburn, MA 01801, USA,), anti-caspase-3 rabbit polyclonal antibody (Catalog no.: A11953, ABclonal, Woburn, MA 01801, USA) and anti-CD8 alpha rabbit monoclonal antibody (Catalog no.: 50389-R309, Sino Biological US Inc. Houston, TX 77074, USA). Sections were washed twice with PBS containing 0.05% Tween-20 (PBS-T) and then incubated with Mouse/Rabbit ImmunoDetector DAB HRP for 1 h at room temperature (Catalog No.: BSB 0003, BIO SB, Santa Barbara, CA 93117, USA.). Finally, slides were washed with PBS-T and incubated with 0.05% DAB and 0.01% H_2_O_2_ for 3 min. The sections were then counterstained with hematoxylin for 1 min, dehydrated in increasing concentrations of ethanol (70%, 80%, 90%, and 100%), and cleared in xylene for 5 min. Finally, all slides were mounted with DPX, cover-slipped, and imaged at 400× magnification using an Olympus light microscope equipped with a digital camera (Olympus, Japan, BX53).

Cells with reaction to insulin, CD8 and caspase-3 antibodies were considered positive. Semi-quantitative analysis of positive-stained tissue sections was performed through modified Allred scoring system guidelines [46]. Positive cells were counted in three pancreatic islets in three different HPFs/mouse (400×) and the mean number was calculated/group.

### 2.3. Statistical Analysis

Statistical analysis was performed with Statistical Package for the Social Sciences (SPSS 21). Differences between the study groups were calculated using repeated measures ANOVA test by the LSD post hoc test. Differences were considered statistically significant at *p* < 0.05.

## 3. Results

### 3.1. Pancreatic Tissues of Mice with Acute and Chronic Toxoplasmosis Demonstrated Marked Insulitis Characterized by Reduced Number and Abnormal Size of Islets of Langerhans

Examination of H & E-stained pancreatic tissue sections of uninfected mice showed uniform, rounded islets of Langerhans within pancreatic acini with no evidence of necrosis or inflammation (Figure 1A, black arrows). In contrast, significant pathological changes were detected in the pancreatic tissue sections of acute and chronic *T. gondii*-infected mice. Mice infected with acute virulent RH strain showed grade 2 insulitis with inflammatory cell infiltration (Figure 1B, black arrows), edema (Figure 1B, arrow heads), and areas of necrosis (Figure 1B, red arrows). Moreover, islets of Langerhans were significantly enlarged compared to uninfected mice (Figure 1B,D; *p* = 0.001). However, pancreatic tissue sections of chronic Me49 strain-infected mice showed grade 1 insulitis and a significant reduction in the size of islets of Langerhans compared to uninfected mice (Figure 1C,D; *p* = 0.001).

Islets of Langerhans were characterized by chronic inflammatory cellular infiltrate (Figure 1C, black arrows) and fibrous-like substance (Figure 1C, red arrow). Interestingly, islets of Langerhans of mice with chronic toxoplasmosis were significantly smaller than those in mice with acute toxoplasmosis (Figure 1D; *p* = 0.003).

Our findings also demonstrate that both *T. gondii*-acute and chronic infections significantly reduced the number of islets of Langerhans compared to uninfected animals (Figure 1E; *p* = 0.018 and 0.021 for acute and chronic infections, respectively). However, there was no statistically significant difference in the number of islets of Langerhans between acute and chronic *T. gondii* infections.

### 3.2. Islets of Langerhans of Mice with Chronic Toxoplasmosis Are Infiltrated with CD8^+^ T Cells

Next, we used immunohistochemical staining to examine infiltration of CD8+ T cells into islets of Langerhans. Our results showed that uninfected (Figure 2A) and acute toxoplasmosis mice (Figure 2B) were negative for CD8^+^ T cell infiltration in islets of Langerhans. On the other hand, islets of Langerhans of mice with chronic toxoplasmosis were infiltrated with CD8^+^ T cells (Figure 2C, black arrows).

### 3.3. Acute and Chronic Toxoplasmosis Are Characterized by Lower Number of β Cells in Islets of Langerhans

Next, we aimed to examine the effect of insulitis on the number of β cells of islets of Langerhans. Immunohistochemical staining of insulin in pancreatic tissue sections revealed higher number of β cells in islets of Langerhans of uninfected mice (Figure 3A, black arrow) compared to *T. gondii*-acute (Figure 3B, black arrow) and chronically (Figure 3C, black arrow) infected mice.

The mean number of β cells were significantly lower in acute and chronic *T. gondii*-infected mice compared to uninfected mice (Figure 3D, *p* = 0.003). In comparison to the chronic toxoplasmosis group, islets of Langerhans of mice with acute toxoplasmosis had statistically significant higher number of β cells (Figure 3D; *p* = 0.021).

### 3.4. Acute and Chronic Toxoplasmosis Are Associated with Apoptotic Cell Death of Islets of Langerhans

Apoptosis of β cells of islets of Langerhans has been previously shown to be associated with the development of T1DM [36]. To examine the apoptotic cell death in the islets of Langerhans in mice with acute and chronic toxoplasmosis, we performed immunohistochemical staining of capsase-3 in pancreatic tissue sections. We detected low levels of caspase-3 in islets of Langerhans of uninfected mice (Figure 4A). Pancreatic tissue sections of mice with acute (Figure 4B) and chronic (Figure 4C) toxoplasmosis showed elevated levels and significantly higher numbers of caspase-3-positive cells (apoptotic cells) in islets of Langerhans (Figure 4D; *p* = 0.001), compared to uninfected animals. Interestingly, the number of caspase-3-positive cells were significantly higher in acute toxoplasmosis than the chronic toxoplasmosis group (Figure 4D; *p* = 0.015).

## 4. Discussion

Diabetes mellitus (DM) is one of the most common endocrine disorders [47]. It has been postulated that several environmental factors including infectious agents can trigger the onset of the disease [27,29]. The causal link between toxoplasmosis and diabetes is still controversial, and they were thought to be predisposed to each other depending on which developed first [48]. Chronic toxoplasmosis was suggested to play a role in the pathogenesis of type 2 diabetes mellitus (T2D) due to the correlation between the state of insulin resistance in T2D and the elevated levels of circulating inflammatory cytokines including IL-2,4; IL-6,5; IL-12,6; TNF; and IFN [48]. Some reports suggested that *T. gondii* infection can cause T1DM due to detection of tachyzoites and bradyzoites in the pancreatic tissues of experimentally infected animals [49]. However, the particular mechanism through which chronic toxoplasmosis could lead to the development of T1D has not yet been elucidated, to our knowledge.

In the present study, we examined the pancreatic tissues of mice infected with acute and chronic *T. gondii* strains to assess the effect of *T. gondii* infection on the pancreas. Significant *T. gondii*-induced pathological changes were detected in both toxoplasmosis groups. The most important finding was insulitis, which was demonstrated in RH and Me49 *T. gondii*-infected animals. It is well documented that insulitis is the main pathological finding in T1DM and individuals who develop insulitis will eventually progress to T1DM [50,51,52,53]. In agreement with our report, *T. gondii*-induced insulitis has been reported by Nassief Beshay et al. [36]. It has been postulated that cytotoxic CD8^+^T cells are the most predominant immune cells that infiltrate islets of Langerhans of pancreas causing insulitis and destruction of the insulin-producing-β cells of islets of Langerhans [50,51]. Given that *T. gondii* is an intracellular pathogen, CD8+ T-lymphocytes, which identify and eliminate intracellular infections in cells infected with viral, bacterial, and parasitic organisms, are anticipated to play a major part in the immune response against this parasite [52]. Our results showed the infiltration of CD8^+^ T cells into the islets of Langerhans in the chronic toxoplasmosis group. This may be explained by the ability of Me49 strain to induce a pro-inflammatory response characterized by exacerbated Th1 response [8]. This finding is consistent with insulitis and autoimmune destruction of β cells of islets of Langerhans in cases of chronic toxoplasmosis, and possible progression to T1DM [53]. On the contrary, the pancreatic tissues of RH infected mice showed no infiltration with CD8+ T cells, which may be explained by that RH strain mostly causing acute death of mice in a short time. Additionally, it has been documented that RH T. gondii dampens the Th1-type immune response [8].

It was interesting that chronic in the toxoplasmosis group showed deposition of fibrous-like material within the islets of Langerhans. The deposition of fibrous-like material has been reported in one study among the pathological alterations induced by streptozotocin, a chemical commonly used in induction of T1DM [54].

It is well documented that reduction of the number and size of the islets are hallmarks of T1DM pathology and this reduction is associated with subsequent reduction of insulin secretion. [55,56]. Our findings showed that number of islets of Langerhans were significantly lower in acute and the chronic toxoplasmosis groups compared to uninfected mice. However, acute toxoplasmosis mice had enlargement islets of Langerhans due to edema and infiltration of leukocytes. On the contrary, the chronic toxoplasmosis group showed marked reduction of the size of the Islets of Langerhans. No statistically significant difference in the number of Islets of Langerhans was observed between the RH and Me49-infected animals. Similar observations were previously reported in pancreatic tissues of Me49 *T.gondii*-infected mice [36]. Moreover, reduced size and number of islets were observed in diabetic mice in which alloxan and streptozotocin were used to induce T1DM [54,57,58,59,60].

Several studies have suggested that apoptosis of β cells of islets of Langerhans is a critical step in the development of T1DM [61]. In line with a study by Nassief Beshay et al. [36], we demonstrated a significantly higher expression of caspase-3 and evident apoptosis in the islets of Langerhans in Me49 *T. gondii*-infected mice compared to the uninfected control group. The role of apoptosis of β cells of islets of Langerhans in induction of T1DM is supported by a study, which showed that caspase-3 deficient mice were protected from acquiring diabetes in a multiple-low-dose streptozotocin autoimmune diabetes paradigm [61].

We found that the number of insulin-producing β cells was significantly lower in the acute and the chronic toxoplasmosis groups (as demonstrated by the lower number of anti-insulin immunoreactive cells) compared to uninfected group. This finding is supported by previous results obtained by Nassief Beshay et al. [36]. Similarly, Ahmadi et al. found a decreased number of insulin immunoreactive cells in the pancreatic islets of diabetic rats [62]. This reduction in number of β cells of islets of Langerhans could be attributed to the observed apoptosis and reduction of the size of the islets.

Based on the results of previous studies, both RH and Me49 *T. gondii* strains have been isolated from human patients [63,64]. So, in the present study we aimed to assess the effect of both strains on the pancreas. However, we do acknowledge some limitations in the present work. The major limitation is the difference between both strains in the route of infection, pathogenicity and the survival of both groups of infected mice and subsequently the difference in follow up times.

## 5. Conclusions

The development of T1DM involves genetic and environmental factors that trigger an autoimmune response that leads to destruction of β cells of islets of Langerhans [65]. In our study, we showed that acute and chronic *T. gondii*-infected mice demonstrated insulitis, reduced number of islets of Langerhans and the insulin-producing β cells, and increased apoptosis of cells within islets of Langerhans. However, unlike acute RH strain-infected mice, islets of Langerhans of chronic Me49 strain-infected mice were infiltrated with CD8^+^ T cells, which were previously associated with chronic toxoplasmosis and development of T1DM [36]. To the best of our knowledge, this is the first study showing the pathological effect of acute RH strain of *T. gondii* on the pancreas, but more studies are needed to explain the mechanism of such changes. We believe that the observed pathological changes in pancreatic tissues of acute and chronic toxoplasmosis mice groups may explain the association of *T. gondii* infections with the development of T1DM.

To further corroborate the link between toxoplasmosis and the development of T1DM, more prospective and retrospective cohort studies are needed on the incidence of T1DM among *T. gondii* seropositive patients. The establishment of implication of *T. gondii* infection in T1DM development could help with diabetes risk prediction, early therapeutic intervention, and potential utility of *T. gondii* therapeutics for prevention of T1DM.

## Figures and Tables

**Figure 1 biomedicines-11-00018-f001:**
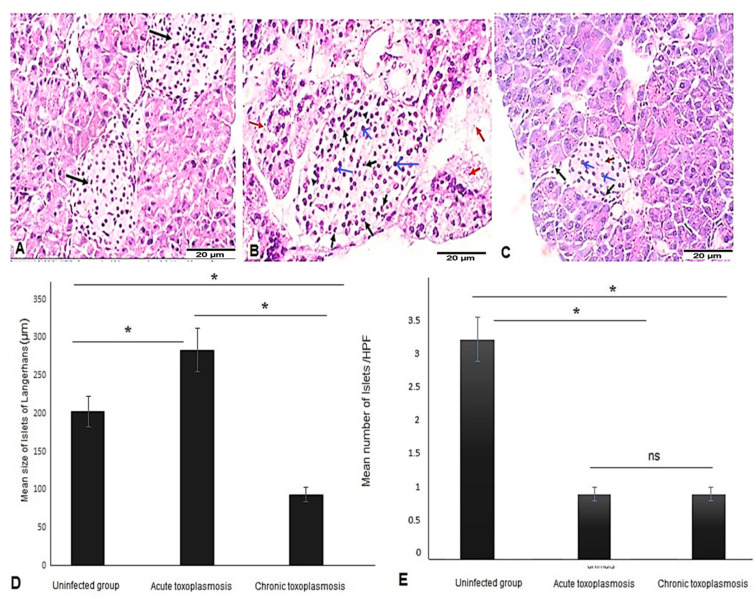
Acute and chronic toxoplasmosis are associated with pancreatic pathological changes and reduced the numbers of islets of Langerhans. Pancreatic tissue sections of mice from different groups (*n* = 10/group) were stained with H & E stain and imaged at 400×. (**A**) Representative image of pancreatic tissue sections of uninfected mice showing uniform rounded islets (black arrows) within pancreatic acini, with no inflammation, edema or necrosis. (**B**) Representative image of pancreatic tissue sections from the acute toxoplasmosis group showing enlarged islets of Langerhans with β cells (blue arrows), acute inflammatory infiltrate (black arrows), edema (arrow heads), and areas of necrosis (red arrows). (**C**) Representative image of pancreatic tissue sections from the chronic toxoplasmosis group showing significant reduction in size of the islets of Langerhans with β cells (blue arrows), mild infiltration by chronic inflammatory cells (black arrows) and areas showing fibrous-like material (red arrow). (**D**) Size of islets of Langerhans in uninfected, acute, and chronic toxoplasmosis mice groups. (**E**) Number of islets of Langerhans in uninfected, acute, and chronic toxoplasmosis mice groups/HPF. Data are expressed as mean ± SD (*n* = 10). Asterisks (*) indicate a statistically significant difference; *p* < 0.05 and “ns” indicates insignificant difference.

**Figure 2 biomedicines-11-00018-f002:**
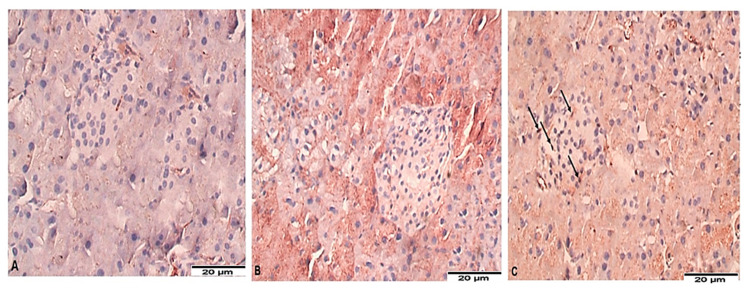
CD8^+^ T cells Infiltrated pancreatic islets of Langerhans of mice with chronic toxoplasmosis. Immunohistochemistry representative images of pancreatic tissue sections stained for CD8 showing absence of CD8^+^ T cell infiltration into islets of Langerhans of uninfected (**A**) and acute toxoplasmosis mice (**B**) but marked infiltration of CD8^+^ T cells into islets of Langerhans of chronic toxoplasmosis mice group (**C**).

**Figure 3 biomedicines-11-00018-f003:**
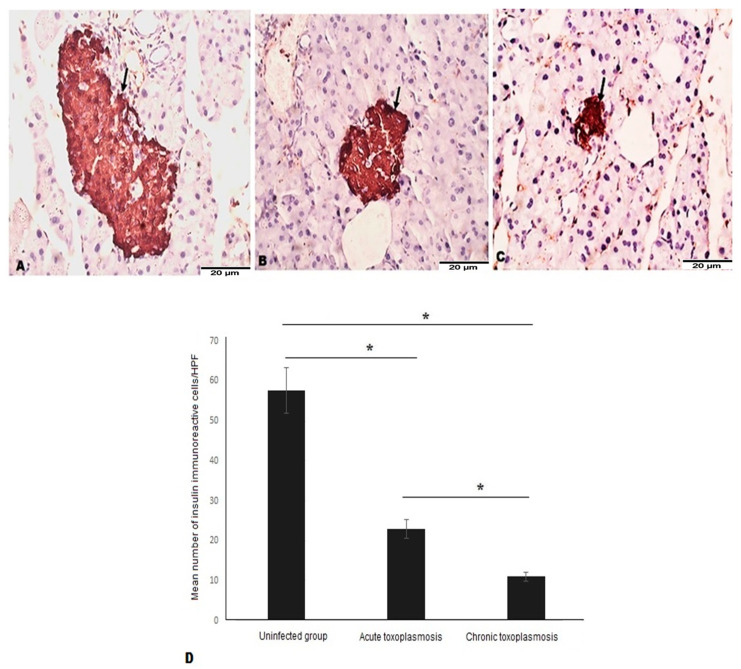
Acute and chronic toxoplasmosis are associated with significantly lower number of insulin-producing-β cells of islets of Langerhans. Pancreatic tissue sections of mice from different groups (*n* = 10/group) were stained with anti-insulin antibody. (**A**) Representative image of pancreatic tissue sections of uninfected mice with strong insulin staining. (**B**) Representative image of pancreatic tissue sections of the acute toxoplasmosis group with smaller stained area of islets of Langerhans (fewer β cells) compared to uninfected group. (**C**) Representative image of pancreatic tissue sections of the chronic toxoplasmosis group with smaller stained area of islets of Langerhans (fewer β cells) compared to the acute toxoplasmosis group. (**D**) Number of β cells (insulin producing cells)/HPF in islets of Langerhans of different mice groups. Data are expressed as mean ± SD (*n* = 10). Asterisks (*) indicate statistically significant difference; *p* < 0.05.

**Figure 4 biomedicines-11-00018-f004:**
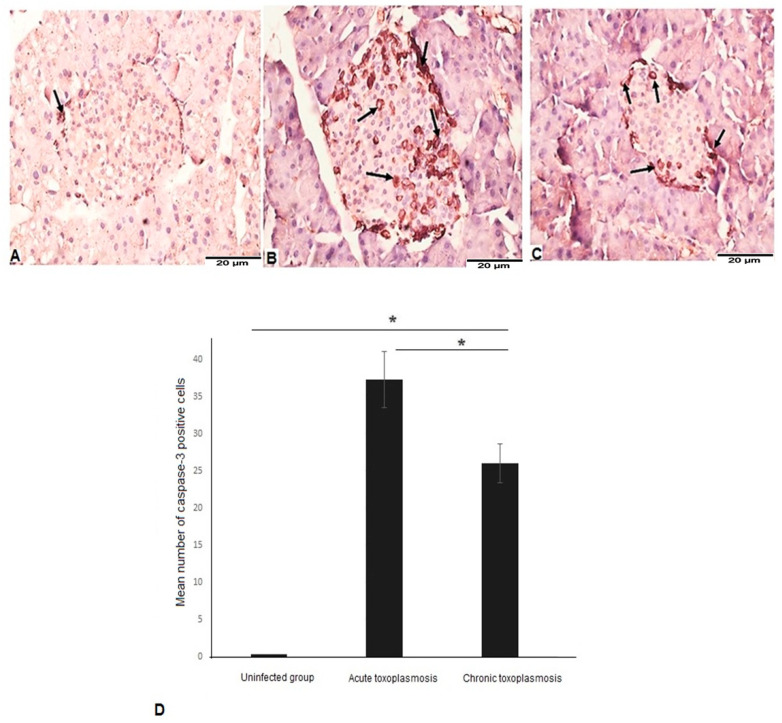
Acute and chronic toxoplasmosis induced apoptotic cell death in pancreatic islets of Langerhans. Pancreatic tissue sections of mice from different groups (*n* = 10/group) were stained with anti-caspase-3 antibody. (**A**) Representative image of pancreatic tissue sections of uninfected mice showing weak caspase 3 signal. (**B**) Representative image of pancreatic tissue sections of the acute toxoplasmosis group showing stronger caspase-3 staining and higher number of caspase-3-postive cells (apoptotic cells). (**C**) Representative image of pancreatic tissue sections of the chronic toxoplasmosis group with high number of caspase-3-postive cells but less than the acute toxoplasmosis group. (**D**) Number of caspase-3 positive cells /HPF in islets of Langerhans of different mice groups. Data are expressed as mean ± SD (*n* = 10). Asterisks (*) indicate statistically significant difference; *p* < 0.05.

## Data Availability

Not applicable.

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
