# Peer review of "Pancreatic Pathological Changes in Murine Toxoplasmosis and Possible Association with Diabetes Mellitus"

_biomedicines, 2022, doi:10.3390/biomedicines11010018_

Round 1

Reviewer 1 Report

The article entitled: ``Pancreatic Pathological Changes Associating Murine Toxoplasmosis and Possible Causal Association between Toxoplasmosis and Diabetes Mellitus`` is an original paper.

The authors assessed the pancreatic pathological changes in mouse model with acute and chronic toxoplasmosis and their association with T1DM. Acute and chronic toxoplasmosis mice displayed marked pancreatic pathological changes with reduced numbers of islets of Langerhans and insulin-producing. However, there was no statistically significant difference in the number of islets of Langerhans between acute and chronic T. gondii infections. Why? Do you have an explanation? Maybe the follow-up was not enough for a chronic toxoplasmosis.

Discussions are very poor. It is difficult to sustain a causal relation between toxoplasmosis and diabetes mellitus. I recommend extending the discussions with the possible mechanisms of these changes.

Author Response

The article entitled: ``Pancreatic Pathological Changes Associating Murine Toxoplasmosis and Possible Causal Association between Toxoplasmosis and Diabetes Mellitus`` is an original paper.

  • The authors assessed the pancreatic pathological changes in mouse model with acute and chronic toxoplasmosis and their association with T1DM. Acute and chronic toxoplasmosis mice displayed marked pancreatic pathological changes with reduced numbers of islets of Langerhans and insulin-producing. However, there was no statistically significant difference in the number of islets of Langerhans between acute and chronic gondii infections. Why? Do you have an explanation? Maybe the follow-up was not enough for a chronic toxoplasmosis.

Response: thank you for your comment. It is well known that the survival rate of mice infected with T. gondii RH strain is much shorter than those infecte with Me49 strain. So the sacrifice of mice was done according to the expected survival according to previous studies. Follow-up for chronic toxoplasmosis was according to previous study by Nassif et al., with similar findings.

  • Discussions are very poor. It is difficult to sustain a causal relation between toxoplasmosis and diabetes mellitus. I recommend extending the discussions with the possible mechanisms of these changes.

Response:  thank you for your comment. Discussions are now improved.

Reviewer 2 Report

In this work, El-Kady et al describe the relationship between Toxoplasma infection and signs related to type 1 diabetes in mice. The manuscript is in general well written, and experiments well performed. However, the manuscript is quite simple with just an experimental infection of mice and the histological study of their pancreas. In addition, and more importantly, it has critical limitations that in my opinion renders it not apt for publication in Biomedicines.

The biggest limitation comes with their biased experimental design of infection. Conclusions drawn from this work rely on the fact that authors compared RH vs ME49 as acute and chronic toxoplasmosis. While it is true that RH causes acute death of mice and ME49 usually causes an eventual chronic infection, using completely different strains that have been widely shown to behave in a completely different way in terms of the elicited immune response, renders results not comparable between them, especially when each strain has been done with different routes of infection and at different time points.

It is broadly described that RH is not cleared by the mouse as it dampens the Th1-type immune response, while ME49 produces a pro-inflammatory response characterized by a usually exacerbated Th1 response (which might, for example account for the increased CD8 presence observed with the ME49 strain). Therefore, because those two strains are not comparable, results between RH and ME49-infected mice in this study at different time points cannot be fully compared to each other. In addition, authors also used different routes for infection for each strain, which might also have an important influence on the immune response.

In general, I feel this work might have been much more conclusive had authors performed a proper comparison using the same Toxoplasma strain and route of infection, checking the same parameters at early and late points after infection in parallel. Alternatively (or even better, additionally), a comparison with different strains but at the same time points and using the same routes of infection, could also have been an interesting approach to test the effect of different strains in the pancreas and its relationship with T1DM. Instead of any of these possibilities, authors did a comparison using 2 different strains with 2 different routes and 2 different time points. Unfortunately, this makes their conclusions unreliable.

Having said this, this work still provides useful information; however, clear conclusions cannot be drawn, and therefore this work should just be made as a descriptive assay and aim at lower journals given its simplicity and biased design. In addition, all the issues mentioned above must be addressed, acknowledged and thoroughly discussed in the manuscript, so readers are aware of the important limitations of the study and that results should be taken with a pinch of salt. Finally, authors did not take into account the fact that, besides the strain type, Toxoplasma also behaves quite differently depending on the host. What has been seen here might not be the same in other hosts, importantly humans. Authors also need to discuss and acknowledge this fact and tone down their assertions.

L329-330: it is incorrect to call the Toxoplasma RH strain “acute” and the ME49 strain “chronic”. A strain is not acute or chronic, a disease is. In fact, ME49 may cause an acute disease shortly after infection, and actually some mice might die because of it. Please, correct this mistake which is present throughout the manuscript.

L337-339: Because authors have not checked whether the ME49 strain, which causes an exacerbated proinflammatory response, could also make CD8 levels increase in the pancreas at early stages of infection, this assertion must be toned down. RH, the strain used for the “acute phase”, is very well known for inhibiting the proinflammatory response in the early stages after infection, possibly explaining why CD8 infiltrates were only seen with the ME49 strain.

Author Response

Thank you for your comments. Below is a point by point response to your comments.

In this work, El-Kady et al describe the relationship between Toxoplasma infection and signs related to type 1 diabetes in mice. The manuscript is in general well written, and experiments well performed. However, the manuscript is quite simple with just an experimental infection of mice and the histological study of their pancreas. In addition, and more importantly, it has critical limitations that in my opinion renders it not apt for publication in Biomedicines.

The biggest limitation comes with their biased experimental design of infection. Conclusions drawn from this work rely on the fact that authors compared RH vs ME49 as acute and chronic toxoplasmosis. While it is true that RH causes acute death of mice and ME49 usually causes an eventual chronic infection, using completely different strains that have been widely shown to behave in a completely different way in terms of the elicited immune response, renders results not comparable between them, especially when each strain has been done with different routes of infection and at different time points.

It is broadly described that RH is not cleared by the mouse as it dampens the Th1-type immune response, while ME49 produces a pro-inflammatory response characterized by a usually exacerbated Th1 response (which might, for example account for the increased CD8 presence observed with the ME49 strain). Therefore, because those two strains are not comparable, results between RH and ME49-infected mice in this study at different time points cannot be fully compared to each other. In addition, authors also used different routes for infection for each strain, which might also have an important influence on the immune response.

In general, I feel this work might have been much more conclusive had authors performed a proper comparison using the same Toxoplasma strain and route of infection, checking the same parameters at early and late points after infection in parallel. Alternatively (or even better, additionally), a comparison with different strains but at the same time points and using the same routes of infection, could also have been an interesting approach to test the effect of different strains in the pancreas and its relationship with T1DM. Instead of any of these possibilities, authors did a comparison using 2 different strains with 2 different routes and 2 different time points. Unfortunately, this makes their conclusions unreliable.

Having said this, this work still provides useful information; however, clear conclusions cannot be drawn, and therefore this work should just be made as a descriptive assay and aim at lower journals given its simplicity and biased design. In addition, all the issues mentioned above must be addressed, acknowledged and thoroughly discussed in the manuscript, so readers are aware of the important limitations of the study and that results should be taken with a pinch of salt. Finally, authors did not take into account the fact that, besides the strain type, Toxoplasma also behaves quite differently depending on the host. What has been seen here might not be the same in other hosts, importantly humans. Authors also need to discuss and acknowledge this fact and tone down their assertions.

Response: Thank you for your comment. the present study is not a comparative study. The main aim of the present work was to assess the effect of 2 strains of toxoplasma - which have been previously reported to infect humans - on the pancreas. Both RH and ME49 T. gondii strains infect humans according to previous reports (PMID: 16333071, PMID: 20521929).  

Additionally, RH and Me49 strains are the only available laboratory T. gondii strains in Egypt. So, it is difficult to obtain strains other than those to compare.  

Third, It is well known that the survival time of T. gondii infected mice is different between RH and ME49 strains. RH Infected untreated mice were reported to have short survival time from 7 to 10 days (PMID 32850486, 34876824, 31673445).

L329-330: it is incorrect to call the Toxoplasma RH strain “acute” and the ME49 strain “chronic”. A strain is not acute or chronic, a disease is. In fact, ME49 may cause an acute disease shortly after infection, and actually some mice might die because of it. Please, correct this mistake which is present throughout the manuscript.

Response: Thank you for your comment. Corrected as advised.

L337-339: Because authors have not checked whether the ME49 strain, which causes an exacerbated proinflammatory response, could also make CD8 levels increase in the pancreas at early stages of infection, this assertion must be toned down. RH, the strain used for the “acute phase”, is very well known for inhibiting the proinflammatory response in the early stages after infection, possibly explaining why CD8 infiltrates were only seen with the ME49 strain.

Response: Thank you for your comment.

Reviewer 3 Report

Interesting study. It appears to corroborate prior studies so I am not sure how much is original. In any case the study is properly done, and the results appear to be believable.

Several issues:

1. Please state that animals for this study were treated humanely per protocols using lab animals.

2. Line 76 - type 2 diabetes is 11% worldwide. Type 1 DM is closer to 1/2 of one percent. Please correct.

3. In METHODS - describe criteria for grade 1, 2, and 3 insulitis.

4. Figures 1 - I see no difference between the nuclei of an inflammatory cell and of an islet cell. Could a larger magnification be shown - perhaps in an appendix?

5. Fig 2 c - same issue. I see no difference in islet cell nuclei and inflammatory cells.

6. In the Abstract you mention glucose levels in the rodents, but you show no such data. Please provide.

Author Response

Interesting study. It appears to corroborate prior studies so I am not sure how much is original. In any case the study is properly done, and the results appear to be believable.

Several issues:

  1. Please state that animals for this study were treated humanely per protocols using lab animals.

      Response: Thank you for your comment. Clarification was added.

  1. Line 76 - type 2 diabetes is 11% worldwide. Type 1 DM is closer to 1/2 of one percent. Please correct.

            Response: Thank you for your comment. Corrected.

  1. In METHODS - describe criteria for grade 1, 2, and 3 insulitis.

            Response: Thank you for your comment. Clarification was added.

  1. Figures 1 - I see no difference between the nuclei of an inflammatory cell and of an islet cell. Could a larger magnification be shown - perhaps in an appendix?

Response: Thank you for your comment. Figures are magnified. Inflammatory cells have clear crescent shaped nucleus

  1. Fig 2 c - same issue. I see no difference in islet cell nuclei and inflammatory cells.

Response: Thank you for your comment. Figures are magnified. Inflammatory cells have clear crescent shaped nucleus

  1. In the Abstract you mention glucose levels in the rodents, but you show no such data. Please provide.

Response: Thank you for your comment. Corrected in the abstract.

Round 2

Reviewer 1 Report

Thank you for responding to my comments/suggestions.

Author Response

Thanks you very much for your comment.  We answered all the reviewer comments in the first round of review and he had no more questions.  

Reviewer 2 Report

Unfortunately, authors do not seem willing to acknowledge all the critical limitations I pointed out in my previous report, nor have they provided an appropriate rebuttal to discuss each issue further. Many of the points I raised are directly ignored in their letter, others are answered vaguely and drifting out of subject, and more importantly only very little and clearly insufficient changes have been made in the new version of the manuscript. For all these reasons, I have not changed my mind nor received arguments from the authors to do so; therefore, I still recommend rejecting this manuscript. I suggest authors reconsider their stand and aim at lower journals provided they address all the concerns I previously pointed out.

Author Response

thank you for your comment.

Reviewer 3 Report

The paper is fine except for 2 points:

INTRO - Type 1 diabetes is in 0.3% of the US population (1.6 million per 330 million US residents). Remove the part of the sentence re the rest of the world. That may be for Type 2 DM.

Once again i do not see the difference between a beta cell making insulin and white blood cell infiltrating the islet. Perhaps include a note in the description under the photomicrographs of what an islet cell looks like and what a white cell looks like. Please note  - I do not question the veracity f your findings. I just do not see differences in cell types

Author Response

Comments and Suggestions for Authors

INTRO - Type 1 diabetes is in 0.3% of the US population (1.6 million per 330 million US residents). Remove the part of the sentence re the rest of the world. That may be for Type 2 DM.

Response: Thank you for your comment. This sentence is now changed as advised.

Once again I do not see the difference between a beta cell making insulin and white blood cell infiltrating the islet. Perhaps include a note in the description under the photomicrographs of what an islet cell looks like and what a white cell looks like. Please note - I do not question the veracity of your findings. I just do not see differences in cell types

Response: Thank you for your comment. a larger magnification of Figure 1 is now used with arrows pointing to beta cells (Blue arrows) and inflammatory cells (black arrows).